# AUV Trajectory Tracking Models and Control Strategies: A Review

**Daoliang Li** [1,2,3,4,5,*] and **Ling Du** [1,2,3,4,5]

1  National Innovation Center for Digital Fishery, College of Information and Electrical Engineering, China Agricultural University, Beijing 100083, China; duling0615@163.com
2  Beijing Engineering and Technology Research Centre for Internet of Things in Agriculture, College of Information and Electrical Engineering, China Agriculture University, Beijing 100083, China
3  China-EU Center for Information and Communication Technologies in Agriculture, College of Information and Electrical Engineering, China Agriculture University, Beijing 100083, China
4  Key Laboratory of Agricultural Information Acquisition Technology, Ministry of Agriculture, College of Information and Electrical Engineering, China Agriculture University, Beijing 100083, China
5  College of Information and Electrical Engineering, China Agricultural University, Beijing 100083, China
*  Correspondence: dliangl@cau.edu.cn

**Abstract:** Autonomous underwater vehicles (AUVs) have been widely used to perform underwater tasks. Due to the environmental disturbances, underactuated problems, system constraints, and system coupling, AUV trajectory tracking control is challenging. Thus, further investigation of dynamic characteristics and trajectory tracking control methods of the AUV motion system will be of great importance to improve underwater task performance. An AUV controller must be able to cope with various challenges with the underwater vehicle, adaptively update the reference model, and overcome unexpected deviations. In order to identify modeling strategies and the best control practices, this paper presents an overview of the main factors of control-oriented models and control strategies for AUVs. In modeling, two fields are considered: (i) models that come from simplifications of Fossen's equations; and (ii) system identification models. For each category, a brief description of the control-oriented modeling strategies is given. In the control field, three relevant aspects are considered: (i) significance of AUV trajectory tracking control, (ii) control strategies; and (iii) control performance. For each aspect, the most important features are explained. Furthermore, in the aspect of control strategies, mathematical modeling study and physical experiment study are introduced in detail. Finally, with the aim of establishing the acceptability of the reported modeling and control techniques, as well as challenges that remain open, a discussion and a case study are presented. The literature review shows the development of new control-oriented models, the research in the estimation of unknown inputs, and the development of more innovative control strategies for AUV trajectory tracking systems are still open problems that must be addressed in the short term.

**Keywords:** autonomous underwater vehicle; trajectory tracking; modeling; control strategies

## 1. Introduction

An underwater vehicle is a semi-autonomous or fully autonomous underwater robot equipped with sensing, decision-making, and execution capabilities [1]. It has been widely used in marine exploration and mapping, underwater pipeline inspection, and scientific and military missions. Traditional underwater vehicles are provided energy through cables, which limits maneuverability [2]. AUVs are normally cruising with self-carried battery and streamline configuration. In addition, AUVs are usually carrier based and have many applications when used in conjunction with airborne equipment and tools [3]. They are developed to complete environmental perception, location, analysis, decision-making, and operative missions autonomously and independently in complicated environments. Therefore, AUV research is a topic of great interest. Recent developments and applications of AUVs are listed in Table 1.

**Table 1.** AUVs and its application in various countries.

| Country | AUV Name | Research Institute | Working Depth (m) | Research Purpose |
|---|---|---|---|---|
| USA | REMUS-6000 | Woods Hole Oceanography Institute | 6000 | Offshore exploration, survey, and automatic sampling [4] |
| | Odyssey | Massachusetts Institute of Technology | 3000 | Scientific investigation and ocean automatic sampling network research [5] |
| | CETUS | Massachusetts Institute of Technology | 4000 | Military torpedo detection search and danger elimination [6] |
| | SAUVIM | University of Hawaii | 6000 | Cable laying and demining |
| China | CR-02 | Shenyang Automation Institute | 6000 | Mineral resources survey and development [7] |
| | CR-01 | Shenyang Automation Institute | 6000 | Pacific Polymetallic Nodules Survey [8] |
| | "Explorer" | Shenyang Automation Institute | 4500 | Marine search and rescue, undersea resource survey [9] |
| Germany | DeePC | STN Company | 4000 | The ice survey [10] |
| UK | AUTOSUB | Southampton Oceanographic Centre | 1600 | Multi-purpose marine survey and surveillance platform [11] |
| France | ALIVE | Cybernextix Company | 3000 | Equipment maintenance and investigation, archaeology and dangerous goods collection [12] |
| Portugal | Delfim | Dynamical systems and ocean roboticsLAB | 4000 | Collection and transmission of marine data [13] |
| Norway | HUGIN1000 | Konsberg | 1000 | Mine search mission [14] |
| | HUGIN3000 | Konsberg | 3000 | Application of fuel cell to AUV [15] |
| Japan | Tri-TON 2 | University of Tokyo, Japan | 2000 | Detect underwater mineral storage [16] |
| Canada | Theseus AUV | ISE research | 2000 | Ice cable laying [17] |

Although AUV has been studied extensively, motion stability and reliability remain complicated due to uncertainties in an underwater environment [18]. To address these problems, path-following and trajectory tracking control methods have been developed to ensure proper motion of AUV [19]. Path-following control ensures the vehicle to reach a destination by following a desired path along Cartesian coordinates. This task involves the separate construction of geometric paths and dynamic allocation, with an emphasis on spatial convergence in dynamics [20]. However, when performing tasks such as maneuvering target tracking, time-sensitive target strikes, or coordinated formations, AUV must have high precision and rapid response capabilities. Trajectory tracking solves the problem by forcing the AUV to track the time-parameterized path [21]. In the trajectory tracking task, the vehicle must arrive at a certain point at a pre-specified time. Therefore, it inherently integrates the distribution of space and time into a whole [22]. However, the trajectory tracking controller is affected by highly nonlinear vehicle dynamics and external time-varying disturbances. These effects make it difficult to measure or estimate in an underwater environment. Therefore, it is extremely challenging to achieve fast and accurate trajectory tracking for AUVs [23]. The control principle of AUV trajectory tracking is shown in Figure 1. AUV trajectory tracking control includes three parts: trajectory planning, user interface, and trajectory tracking control. The trajectory planning includes path planning, behavior decision, and trajectory generation. It generates the required database, such as task and mode data, vehicle component, and user profile. The GUI displays two aspects of interactive data, including trajectory planning and trajectory control. AUV obtains the planned route through the GUI, and it generates deviation with the actual movement data, and the motion control is achieved through thruster control and energy control, thereby the trajectory tracking error is reduced.

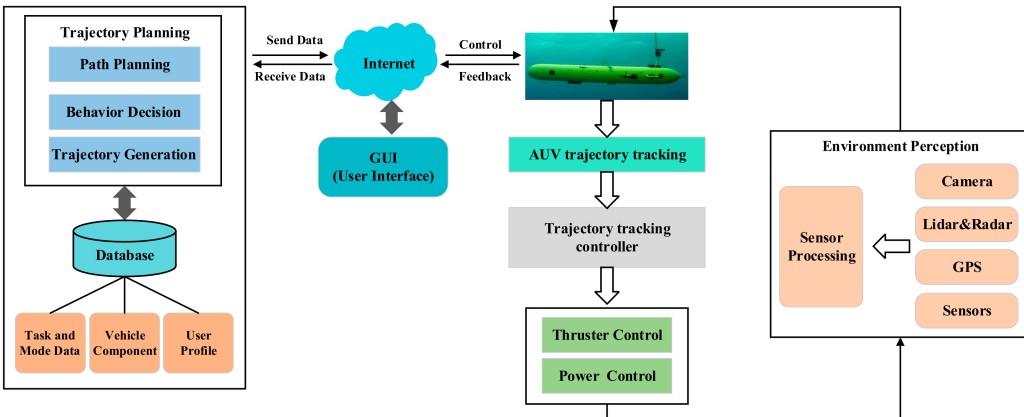

**Figure 1.** AUV trajectory tracking structure block diagram.

The establishment of the trajectory tracking model is helpful to improve the decision-making and intervention ability of the actual system [24]. Generally, two factors need to be considered when establishing an AUV trajectory tracking model: (i) the complexity of the model will affect the calculation efficiency, and (ii) the accuracy of the model is the basis for an accurate analysis of the system [25]. Inspired by robot motion modeling, Fossen developed a six degree-of-freedom (DOF) motion model of a maritime vehicle in vector form [26]. This form can make it easier for researchers to understand the physical meaning of each part of the motion model, and greatly reduce the complexity of AUV motion controller derivation [27]. Nevertheless, due to time-varying external interference, this modeling method is difficult to accurately establish a system model. System identification is based on the determined model structure and uses optimization search such as least squares and genetic algorithms to identify the model parameters [28]. It not only uses various offline data during the AUV operation process, but it also identifies the system parameters online by designing various types of observers, laying the foundation for the accurate establishment of AUV model [29]. Research shows that the system parameter identification method based on neural network (NN) has achieved good performance in AUV trajectory tracking modeling.

Accurate trajectory tracking requires the coordinated control of altitude and position under the dual constraints of time and space [30]. Complicated non-linearity, underactuated system, system constraints, and other issues make AUV trajectory tracking control a challenge [31]. Various approaches have been undertaken to ensure AUV trajectory tracking convergence speed and accuracy. Complex unknown external disturbance is an important issue of trajectory tracking, which could affect the model accuracy and degrade the closed-loop system performance. The commonly used robust control methods include, but are not limited to, sliding mode control, H-infinity control, model predictive control (MPC), NN control, optimal control, multi-agent, etc. [32]. However, conventional controllers with fixed gains fail to guarantee high quality responses of the overall system when significant changes occur in the vehicle dynamics and its environment [33]. Intelligent adaptive control has proven to be successful in several nonlinear applications [34]. In addition, adaptive control provides an ability to re-adjust the controller parameters online to achieve the required performance when the process parameters are unknown and vary over time [35]. In contrast to the motion control of fully actuated AUVs, the main concern in the design of controllers for underactuated AUVs is that the number of their independent actuators is fewer than degrees of freedom [36]. This feature increases the degree of complexity in the design of nonlinear tracking controllers for such systems. In order to overcome some of the above-mentioned problems, several tracking controllers were developed based on the Lyapunov direct method, feedback control law, and the backstepping method [37]. In addition, the system constraints, such as thrust limit and safe operating area, are inevitable in real AUV applications. To include these constraints in the controller designs, MPC is an ideal tool, because it can handle constraints through optimization procedures [38].

Most of the above studies are based on the assumption that all the motion states of AUV are measurable. However, sensor failures may occur to AUVs considering the complex underwater environment [39]. In these cases, state observer-based control methods can provide effective solutions for such conditions. Studies show that the development of new control-oriented models, the research in the estimation of unknown inputs, and the development of more innovative control strategies for AUV trajectory tracking systems are still open problems that must be addressed in the short term.

To address these issues, this review aims to: (a) provide a comprehensive survey and review on the current control models and their associated parameters, algorithms, and strategies; (b) help the AUV investors and developers to determine the appropriate AUV model and control algorithms; (c) identity the future development path for AUV and conduct in-depth research. The rest of the paper is organized as follows. Section 2 describes the modeling method of AUV trajectory tracking control, Section 3 presents the AUV trajectory tracking control strategy, and Section 4 gives the conclusions and future perspectives.

## 2. AUV Trajectory Tracking Model

The AUV spatial motion model is the basis for the analysis of motion characteristics and the research of control algorithms. When building an AUV motion model, it is necessary to consider the motion, the influence of environmental parameters, and the coupling factors. There are two commonly used methods to develop AUV trajectory tracking models: analytical model based and system identification based.

### 2.1. Analytical Model

Analytical model is an abstract description of AUV motion system [40]. It is a multi-factor complex system, comprehensively considering the external environment disturbance and its coupling factors [41]. Due to the influence of uncertain factors, the motion analytical model of AUV is usually a nonlinear system [25]. Considering the above factors and the AUV movement mode, the researchers established a kinematic and dynamic model. The schematic diagram of the AUV motion model is shown in Figure 2. This section mainly reviews the establishment of analytical models from the perspective of analytical modeling methods, factors that affect model establishment, such as system coupling, nonlinearity, and external environmental disturbances.

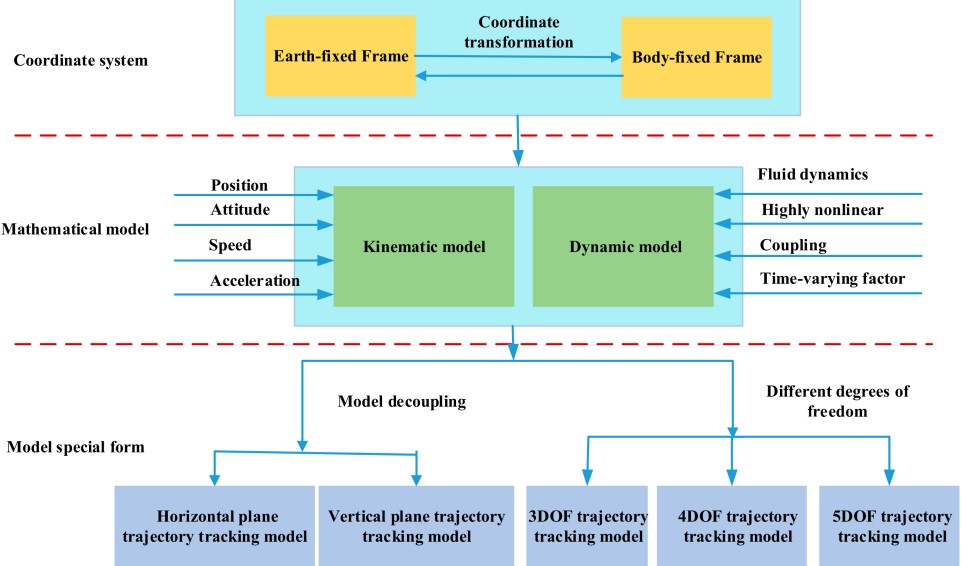

**Figure 2.** A block diagram of a model building system based on dynamic equations.

### 2.1.1. Model Building

AUV motion is generally described in six DOF, which includes translation and rotation components. The translation part includes surge, sway, and heave, which describe the position of the AUV. The rotation components include roll, pitch, and yaw, which describe the direction of the AUV [42]. The description of AUV motion is able to be achieved through kinematics and dynamic models in the body coordinate and inertial coordinate.

Inspired by the robot motion model and vector rigid body dynamics, Fossen proposed a vector form of marine vehicle motion model [43]. This type of dynamic system typically uses the Newton–Euler equation and the Quasi-Lagrange equation to model the dynamics [44], which makes the full use of the physical characteristics of rigid three-dimensional space motion. This model can not only reduce the number of model parameters, but also help to analyze the impact of dynamic equations on the vehicle motion [45]. To simplify analysis and controller design, ref. [46] studied the dynamic equations established by the Quasi-Lagrangian equation and divided the model into a series of interrelated subsystems.

Ref. [47] introduced additional items $\Delta(v, \eta)$ in the six DOF mathematical model, which represents structural or non-structural parameter errors, system disturbances, or unmodeled dynamics, effectively avoiding the impact of modeling errors on AUV performance.

Meanwhile, since the number of underactuated control inputs is always less than the number of state variables, the difference between the level of the input matrix and the dimension of the configuration vector is used to distinguish between a fully actuated and an underactuated AUV. The AUV definitions regarding the fully actuated system, the underactuated system, and the degree of underactuation are listed [48]:

**Definition 1.** *(Fully actuated system): If instantaneous acceleration could be achieved in any direction of v, the system is a fully actuated system. This may also mean that rank $\{R(v)\} = dim(v)$.*

**Definition 2.** *(Underactuated system): If instantaneous acceleration could not be achieved in any direction of v, the system is an underactuated system. This may also mean that rank $\{R(v)\} < dim(v)$.*

**Definition 3.** *(Degree of underactuation): The degree of underactuated is the number of configurations that cannot be controlled immediately. Expressed as: rank $dim(v) - \{R(v)\}$.*

### 2.1.2. System Coupling Factors

The AUV movement process contains six DOF, and there is a strong coupling relationship between each degree of freedom, which makes the design of the controller very complicated [49]. Model decoupling, which can reduce the complexity of the system and shorten the calculation time required by the controller, is of great significance. One option to decompress a six DOF model is to decompress the model into three independent, non-interacting inserts: steering, diving, and speed control. Each subsystem can be linearized at a constant operating point, thereby deriving the control law of each single input multiple output (SIMO) subsystem [50]. Ref. [51] adopted a distributed implementation in which the original optimization problem was appropriately decomposed into three smaller sub-problems. Another decoupling method was usually to ignore the coupling between the rolling surface motion and the two plane motions. In this case, the vehicle motion is divided into vertical motion and horizontal motion [52].

However, when these decoupling methods are applied to highly coupled nonlinear models, the results obtained are not satisfactory. Due to decoupling, the system assumes that the respective degrees of influence do not affect each other, causing conflicts in control inputs. It can be seen from the comparison that the use of the coupling system can improve the trajectory tracking accuracy. The trajectory tracking model of the nonlinear fully coupled system is difficult to analyze, and the output of the affine nonlinear system is linear with respect to the control signal [53]. Therefore, ordinary feedback linearization can be used for these systems.

### 2.1.3. System Nonlinearity Factors

In some work, the highly nonlinear problem of the model can be dealt with by linearizing the model around the operating point. Some researchers reduce the complexity of the AUV model by linearizing its running speed [54]. However, most of the existing linearization methods aim to develop a fully linear model that does not retain any nonlinearity. Therefore, because high-order terms are omitted, there are inevitably unmodeled dynamics. Inaccurate modeling will reduce the system performance [55]. State-dependent Riccati equation control (SDRE) technology can synthesize nonlinear feedback control by allowing the nonlinearity in the system state, and at the same time, it can provide great design flexibility for the control system design of nonlinear dynamic system, and avoid the error caused by traditional linearization treatment [56]. In addition, ref. [55] developed a local compact form dynamic linearization (local-CFDL) to transform the original nonlinear nonaffine system into an affine structure consisting of both an unknown residual nonlinear time-varying term and a linearly parametric term affine to the control input. The local-CFDL model can be rewritten in a compact form:

$$\Delta y(t+1) = \varnothing(t)\Delta u(t) + \xi(t) \tag{1}$$

where $t$ represents the AUV running time, $\Delta y(t+1) = y(t+1) - y(t)$ represents the output increment at the next moment, $\Delta u(t) = u(t) - u(t-1)$ represents the input increment at this moment, $\varnothing(t)$ is presented to denote the partial derivative of unknown nonlinear scalar function $f(\cdot)$ with respect to control input, $\xi(t)$ denotes the residual nonlinear uncertainties of the affined linear data model and will be estimated as a whole in the following controller design process.

### 2.1.4. Environmental Disturbance Factors

The external interference of AUV is very complicated. When the AUV is operating in a very shallow water/surface area, significant interference due to shallow water waves will be introduced into the translational motion of the AUV. Since small AUVs are more sensitive to wave interference, mathematical model of these interferences should be carried out to facilitate motion planning and control purposes [57]. The common solution is to applying a superposition of multiple regular blogs to implement the description of random waves [57]. During the deep dive, the changes in diving depth, pressure, salinity, and density can affect the buoyancy and should be included in the model establish [58]. Depth control of AUVs during vertical plane motion model is written in (2), which includes fluctuation and nonlinearity, and the linear term and nonlinear term are separated.

$$\begin{cases} \dot{x}(t) = \begin{bmatrix} A_1 & 0_{2\times2} \\ A_2 & A_3 \end{bmatrix} x_t + \begin{bmatrix} B_1 \\ 0_{2\times1} \end{bmatrix} u(t) + \begin{bmatrix} D_1 \\ 0_{2\times2} \end{bmatrix} v(t) + \begin{bmatrix} f_1 \\ f_2 \end{bmatrix} = Ax(t) + Bu(t) + Dv(t) + f(x,t) \\ x(t_0) = x_0 \end{cases} \tag{2}$$

where the state vector as $x(t) = [w\,q\,\theta\,z]^T$, the wave disturbances vector as $v(t) = [Z_{wave}\,W_{wave}]^T$, the control vector as $u(t) = \delta_s(t)$, and the nonlinear vector as $(x,t) = [f_1\,f_2]^T$.

### 2.2. System Identification

The highly nonlinear and cross-coupled system dynamics, together with the unpredictable complex underwater environment which introduces considerable disturbances and uncertainties, challenges the model establish [59]. System identification-based models focus on developing recognition algorithms to accurately estimate unknown parameters of a model. Thus, the mathematical model could be accurately constructed to achieve more accurate design [60]. System identification methods are divided into offline and online methods based on the data source [61]. The schematic diagram of AUV system identification is depicted in Figure 3.

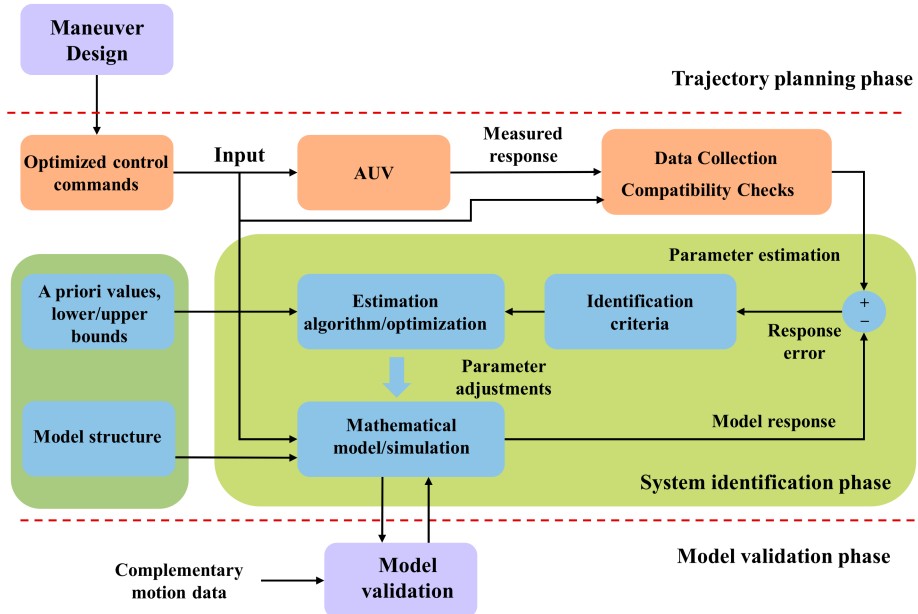

**Figure 3.** The schematic diagram of AUV system identification [62].

### 2.2.1. Offline Identification

Offline system estimation techniques tend to rely on repetitive methods, where data are collected by the vehicle's sensors, then filtered and processed. The most common offline system identification method is the least squares (LS) [63]. The LS linear system acts as the loss function and the system of solution equations. Its level of precision is comparable to the identification method of nonlinear dynamic systems and has the ability of learning fast [64]. Ref. [63] designed a Nonlinear Auto-Regressive Moving Average eXogenous (NARMAX) model. The parameters of the NARMAX model are updated using a recursive extended least square algorithm at each time instant. However, the interference of various environmental factors during the operation of AUV makes the offline measurement data noisy, and at the same time, leads to the uncertainty of the system's state [65]. To build a more accurate model, it is necessary to apply filtering techniques [66]. Commonly used filtering methods include random filters and Kalman filters (KFs). The random filtering method is based on a two-step Bayesian process, that includes time or measurement updates [67]. The KF assumes uncertainty in the dynamics of the Gaussian distribution system and uses the mean and covariance of the state vector for update adjustments [68]. Observer KF was proposed to handle measurement noise and mild nonlinearity. However, the AUV system is influenced by nonlinear factors and require nonlinear filtering methods [69]. Extended Kalman filter (EKF) is commonly used in underwater environments to implement nonlinear filtering [42], which uses the instantaneous linearization of each time step to approximate nonlinearity [70].

### 2.2.2. Offline Identification

The online system identification method collects data and estimates system dynamics as the vehicle operates in real time [71]. This allows the model parameters to update automatically and more reliably, especially when changes to the environmental conditions occur [72]. In the field of online identification techniques, NN, and KF techniques have been investigated for decoupled and coupled motion models. NN has an inherent capability of approximating nonlinear functions for AUV depth trajectory tracking control. Ref. [73] used NN approximators and adaptive robust control strategies to estimate model uncertainties in the design process due to unknown vehicle parameters, unmodeled dynamics, and constant or time-varying interference caused by waves and ocean currents. Ref. [74] used linear parameterized NN (LPNN) to estimate the uncertainty of vehicle dynamics, in which the basis function vector of the network is constructed according to the physical

characteristics of the vehicle. Ref. [58] adopted the NN to approximate the complex AUV hydrodynamics and differential of desired tracking velocities. The bound of the generalized disturbance, which is composed of NN approximation error and ocean disturbances, are approximated based on the adaptive estimation technique. At the same time, the KF estmates optimal states for a linear system using Gaussian error statistics [75]. Based on KF recognition, ref. [76] established a set of decoupled AUV subsystems with different degrees of freedom. In addition, the extended Kalman filter (EKF) is another effective tool for AUV system identification. The EKF is a recursive updating method that has been widely used in AUV system identification for its excellent performance in arithmetic robustness, recognition accuracy, and fast convergence [42].

The online system identification methods are more reliable when the AUV parameters and environmental conditions vary. However, this modeling method lacks transparency and has heavy computation burden due to the large dataset. The multi-model framework realizes modeling and identification of complex nonlinear systems through problem decomposition. The global system model consists of a set of models that integrate different degrees of effectiveness. The multi-model framework can directly and qualitatively integrate object knowledge, which is simpler than the system identification method [61]. The comparison of different modeling methods in AUV trajectory tracking is shown in Table 2.

**Table 2.** Summary of the research studies on the modeling of AUV trajectory tracking system.

| Classification | Authors | Influencing Factors | Method of Modeling | Important Finding |
|---|---|---|---|---|
| Analytical model | [43] | The external environment disturbance and its coupling factors. | The Newton–Euler equation and the Quasi-Lagrange equation. | Not only reduces the number of model parameters, but also helps to analyze the impact of dynamic equations on the vehicle motion. |
| | [46] | The external environment disturbance and its coupling factors. | Quasi-Lagrangian equation. | Divided the model into a series of interrelated subsystems. |
| | [47] | The external environment disturbance and its coupling factors. | Introduces additional items $\Delta(v, \eta)$. | Avoid nonlinear modeling errors. |
| | [51] | System coupling. | The optimization problem can be broken down into three smaller sub-problems. | This sub-problem should be solved in parallel so the calculation time can be greatly reduced. |
| | [77] | System coupling. | Ignore the coupling between the rolling surface motion and the two plane motions. | Shorten the calculation time required to determine each controller. |
| | [53] | System coupling. | Affine nonlinear systems. | Improved trajectory tracking accuracy |
| | [54] | Complexity and the nonlinearity of the model. | Linearizing an operating forward speed. | The high nonlinearity of the model is handled. |
| | [56] | Complexity and the nonlinearity of the model. | SDRE | Offer great design flexibility systematic and effective means for the design of control systems for nonlinear dynamical systems. |

| Classification | Authors | Influencing Factors | Method of Modeling | Important Finding |
|---|---|---|---|---|
| Analytical model | [1] | Complexity and the nonlinearity of the model | local-CFDL | Avoid errors caused by the traditional linearization process. |
| | [57] | The significant interference caused by shallow water waves is introduced into the translational motion of the AUV. | A superposition of multiple regular blogs to implement the description of random waves. | Perform mathematical model of these disturbances to facilitate exercise planning and control purposes. |
| System identification | [64] | N/A | least squares (LS) | Offline system identification method. |
| | [69] | Measurement noise and mild nonlinearity. | Observer Kalman filter. | Remove noise and nonlinearity to make the model more accurate. |
| | [42] | The AUV system is influenced by nonlinear factors and require nonlinear filtering methods. | Extended Kalman filter (EKF). | Use the instantaneous linearization of each time step to approximate nonlinearity. |
| | [73] | The external environment disturbance and its coupling factors. | NN approximator and adaptive technology. | Estimate the uncertainty of the model due to unknown vehicle parameters, unmodeled dynamics, and constant or time-varying disturbances caused by waves and ocean currents. |

## 3. AUV Trajectory Tracking Control

### 3.1. Significance of Control Strategy in AUV Trajectory Tracking

AUV trajectory tracking requires the vehicle to track a time-varying reference trajectory in space. It not only requires the center of mass of the AUV to move along the trajectory, but also requires the arrival time and even the velocity and attitude of the arrival. Due to the complexity of the external environment and its own structure, AUV trajectory tracking control mainly includes the following aspects: (1) Nonlinearity is a prominent feature of AUV in terms of dynamics. Therefore, the obtained dynamic coefficients are difficult to be accurate, and ocean currents are more likely to interfere with it. Based on the above reasons, AUV puts forward higher requirements for its own robustness, adaptive or self-tuning controllers are essential; (2) due to the influence of the uncertainty of the AUV underactuated system, it is necessary to focus on its nonlinear robust control, which is also the difficulty of developing AUV; (3) ignoring system constraints during the design phase may result in degraded trajectory tracking performance. Therefore, it is desirable to consider these constraints in the design of the tracking controller.

### 3.2. Methodologies of Control Strategies Applied in AUV Trajectory Tracking

3.2.1. Mathematical Modeling Study

Many mathematical models of control strategies for AUV trajectory tracking have been studied in the literature. This section will introduce several typical control mathematical models.

Optimal Control

(1)    Linear quadratic regulator (LQR)

The LQR is a highly effective method to design an optimal full-state-feedback controller for linear, or linearized systems [78]. The correction and design of the LQR needs to find the appropriate state variables and control quantity weighting matrix according to

the response curve, without determining the closed-loop pole position according to the required performance [79].

Ref. [80] designed an LQR based on a linearized model and applied it to a nonlinear model to track the desired trajectory. Consider the linear time-varying state space model:

$$\dot{\tilde{\chi}} = At\tilde{\chi}(t) + B(t)\tilde{U}(t) \tag{3}$$

where $\tilde{\chi} = \chi - \chi_{eq}$ and $\tilde{U} = U - U_{eq}$, $A(t)$ is a 12 × 12 matrix and $B(t)$ is a 12 × 4 matrix, and they are calculated as follows: $A(t) = \left.\frac{\partial_F}{\partial_\chi}\right|_{\substack{\chi = \tilde{\chi}_{eq}(t) \\ U = \tilde{U}_{eq}(t)}}$, $B(t) = \left.\frac{\partial_F}{\partial_u}\right|_{\substack{\chi = \tilde{\chi}_{eq}(t) \\ U = \tilde{U}_{eq}(t)}}$.

The feedback LQR control law can be written as follows:

$$U(t) = -K(t)\tilde{\chi}(t) \tag{4}$$

where $U(t)$ represents a vector of feedback control input of length 4, $\tilde{\chi}(t) = \chi(t) - \chi_{eq}(t)$ represents a state error vector of length 12, and $K(\text{t})$ represents a 4 × 12 matrix that contains control gains that vary over time.

However, traditional LQR algorithm does not consider current disturbances and AUV dynamics. To simulate the actual situation, the excitation must be known before determining the optimal control force to obtain a more reliable solution [81].

(2)   State-dependent Riccati equation (SDRE) control

The SDRE control solves the optimal control problem of nonlinear systems by constructing a linear structure, which has great flexibility and ensures a large range of progressive stability through the selection of the State-dependent coefficient matrix weight coefficient [82].

Ref. [56] obtained the nonlinear feedback control law through the quasi-linearization of the dynamic equation and the calculation of the algebraic Riccati equation. Avoid errors caused by traditional linearization processing. State-dependent coefficient (SDC) of the nonlinear dynamic equations specified as:

$$\dot{x} = Ax + Bu \tag{5}$$

where, $\mathbf{x} = [\xi\ \zeta\ u\ w\ q\ x_s\ x_b]^T$, $\mathbf{u} = [T\ u_s\ u_b]^T$, $A = H^{-1}F$, $B = H^{-1}E$, $H_{12} = \begin{bmatrix} H_{11} & H_{12} \\ H_{21} & H_{22} \end{bmatrix}$, $F = \begin{bmatrix} F_{11} & F_{12} \\ F_{21} & F_{22} \end{bmatrix}$, $E = \begin{bmatrix} E_{11} \\ E_{21} \end{bmatrix}$.

Ref. [53] used an SDRE controller in a non-affine structure without decoupling the six DOF. The position error of 43.6 mm is negligible compared the with total measured distance of 8.77 m. The control law is in the form of:

$$u(t) = -R^{-1}\frac{\partial\}(x(t), u(t))}{\partial u(t)}k(x(t), u(t))x(t) \tag{6}$$

where $k(x(t), u(t))$ is the symmetric positive definite solution of the State-dependent Riccati equation as:

$$k(x(t), u(t))A(x(t)) + A^T(x(t)k(x(t), u(t)) - k(x(t), u(t))B(x(t), u(t))R^{-1}B^T(x(t), u(t))k(x(t), u(t)) + Q = 0 \tag{7}$$

where $Q$ is the weighting matrix for states, symmetric positive semi definite, and $R$ is weighting matrix for the control inputs, $x(t) \in \Re^n$ is the state vector, $u(t) \in \Re^m$ is the subsystem speed, $A(x(t)) : \Re^n \to \Re^{n \times n}$ and $B(x(t), u(t)) : \Re^n \to \Re^{n \times n}$, prepares the system for implementing the suboptimal control.

Ref. [83] optimized the system by using the quadratic performance index to solve the SDRE and obtain the sub-optimal control law of the input unconstrained model. Al-

though the system parameters and control fin deflection constraint conditions are uncertain, effective depth control can still be achieved. This control law is given by:

$$\dot{x}_{a1} = \begin{bmatrix} A(x,p) & 0_{4\times1} \\ C & 0 \end{bmatrix} \begin{bmatrix} x \\ x_{s1} \end{bmatrix} + \begin{bmatrix} B(p) \\ 0 \end{bmatrix} \delta_s + \begin{bmatrix} d \\ -Z_r \end{bmatrix}$$

$$\triangleq A_{a1}(x,p)x_{a1} + B_{a1}(p)\delta_s + Ev$$

(8)

where 0 denotes null matrices of appropriate dimensions, introducing the dependence of matrices $A_i$ and $B_1$ on the perturbation vector $p$, $B(P) = \left[B_1{}^T(P), 0^T\right]^T \in R^4$ and $d = \left[d_1{}^T, 0^T\right]^T \in R^4$, $Z_r$ is prescribed depth, $x_{s1}$ is the integral of the depth trajectory tracking error, $x_{a1}$ is the state, $\delta_s$ is input, $C = [0,0,1,0]$, $v = \begin{pmatrix} d_1 \\ Z_r \end{pmatrix} \in \Omega_v \subset R^3$,

$$E = \begin{bmatrix} 1 & 0 & 0 \\ 0 & 1 & 0 \\ 0 & 0 & 0 \\ 0 & 0 & 0 \\ 0 & 0 & -1 \end{bmatrix}.$$

It can be seen from the above research that SDRE shows good effects in system coupling and nonlinear processing. However, there are situations where global asymptotic stability could not be achieved. In this case, estimating the attractive area is valuable and essential. More important, a method of estimating the size of the attractive area could be used.

(3)　Model Predictive Control (MPC)

MPC recursively solves the open-loop optimal control problem by using real-time state measurement as the initial condition [84]. It systematically incorporates the system state and controls the input constraints [26]. Due to the high flexibility in expressing various control problems, it allows MPC to digest any nonlinearity of the system model without any approximations.

Ref. [39] developed a disturbance observer-based nonlinear MPC scheme for cross tracking of underactuated AUV under sea current disturbance. The stability time is 110 s, and the error is under the 5% limit. The proposed model predictive control system with disturbance observer is shown in Figure 4.

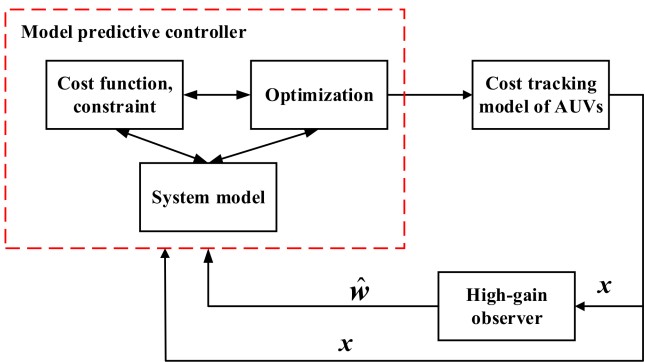

**Figure 4.** Nonlinear model predictive control system with disturbance observer (Reprinted from Reference [39] with permission from IJMS, copyright 2017).

Ref. [59] developed a Lyapunov-based nonlinear MPC (LMPC). Linearization can be avoided by adding shrinkage constraints in MPC and the closed-loop stability of the controller can be ensured. The formula of LMPC can be expressed as Equation (9).

$$\begin{matrix} min \\ \hat{u} \in S(\delta) \end{matrix} \quad J = \int_0^T \|\tilde{X}(S)\|_Q^2 + \|\tilde{u}(S)\|_R^2 ds + \|\tilde{X}(T)\|_P^2$$

(9)

where $\widetilde{X}(S)$ is the predicted state trajectory of the vehicle with respect to the predictive control $\widetilde{u}(S)$, evolving from $X(t_0)$ using the system model, $\widetilde{X} = \hat{X} - X_d$ is the error state and $\widetilde{u} = \hat{u} - u_d$ is the control error, $S(\delta)$ denotes the family of piecewise constant functions characterized by the sampling period $\delta$ and $T = N\delta T$ is the prediction horizon, the weighting matrices $Q$, $R$, and $P$ are positive definite.

Considering the high computational complexity of nonlinear programming related to NMPC, ref. [51] proposed the distributed MPC (DMPC) implementation to alleviate the computational burden. The average computation time confirms the improvement with the DMPC implementation. In addition, NMPC faces a fierce conflict between a longer calculation time and a shorter system sampling period. To address this conflict, ref. [85] modified the continuation/generalized minimal residual (C/GMRES) algorithm. The C/GMRES algorithm efficiently solves the NMPC problem within 10% of the sampling period.

During the underwater operation of AUV, there are inevitably problems such as input saturation and state constraints. Ref. [86] designed the NMPC scheme to use ocean currents for the calculated control input to keep the energy consumed by the thruster at a reduced level. Various constraints were considered during this period, such as sparse obstacles, working space boundaries, control input saturation, and predetermined vehicle speed limits.

Although MPC has achieved good performance, it has the disadvantage of involving complex calculations. Nonlinear MPCs face fierce conflicts between longer calculation times and shorter system sampling periods. Therefore, according to the real-time requirements of MPC, different strategies including offline precomputation, delay compensation, event-triggered strategy, and numerical continuation are proposed in an attempt to shorten the calculation time. The comparison between different optimal control methods in AUV trajectory tracking is shown in Table 3.

**Table 3.** Optimal control method in AUV trajectory tracking.

| Control Strategy | Classification | Improvement | Control Object | Control Effect | Ref. |
|---|---|---|---|---|---|
| LQR | LQR | — | Track the reference trajectory. | Accurate tracking of spiral, sawtooth paths and 3D Dubin paths. | [80] |
| SDRE | SDRE | A hyperbolic tangent sigmoid function is introduced to equivalently replace the rudder angle variable. | Achieve the rudder saturation constraint problem. | The error of the trajectory tracking to converge smoothly to the steady state value. | [56] |
| | | The quadratic performance index. | Deal with suboptimal underwater surface control problems in AUV. | Depth control is achieved with actuator saturation and parameter uncertainty. | [83] |
| MPC | MPC | Genetic algorithm. | Track the reference trajectory. | Track the given nonlinear path with satisfactory accuracy. | [87] |
| | | Recurrent neural network. | Control of AUVs in a vertical plane. | Track the given nonlinear path with satisfactory accuracy. | [88] |
| | LMPC | C/GMRES algorithm. | Handling the actual constraints of the AUV thruster. | Improve the algorithmic efficiency of the NMPC algorithm. | [85] |
| | DMPC | Subproblems and the warm start strategy. | Solve AUV control problems to track time-varying trajectories. | Reduce controller runtime and achieve good control. | [51] |

Nonlinear Time-Invariant Control

(1)    Sliding mode control (SMC)

A remarkable trait of the SMC is its robustness to time-varying parameters, and external environment interference [77]. In addition, terminal sliding mode control (TSMC) has a better effect in convergence speed, interference suppression ability, and uncertainty problems.

To achieve fast convergence and high steady-state tracking accuracy, TSMC is considered to be superior to traditional SMC techniques. Ref. [89] constructed a robust disturbance rejection control law using disturbance observers and modified TSMC. Lyapunov analysis is performed to prove stability and performance. However, the problem with TSMC is the singularity problem with unbounded control inputs [90]. To solve the singularity problem, ref. [91] proposed an adaptive non-singular integral terminal sliding mode control (ANITSMC), which makes the speed and position tracking error locally converge to zero within a finite time. The results show that the ANITSMC can achieve faster convergence speed and better anti-interference effect than adaptive proportional integral sliding mode control. ANITSMC control block diagram shown in Figure 5.

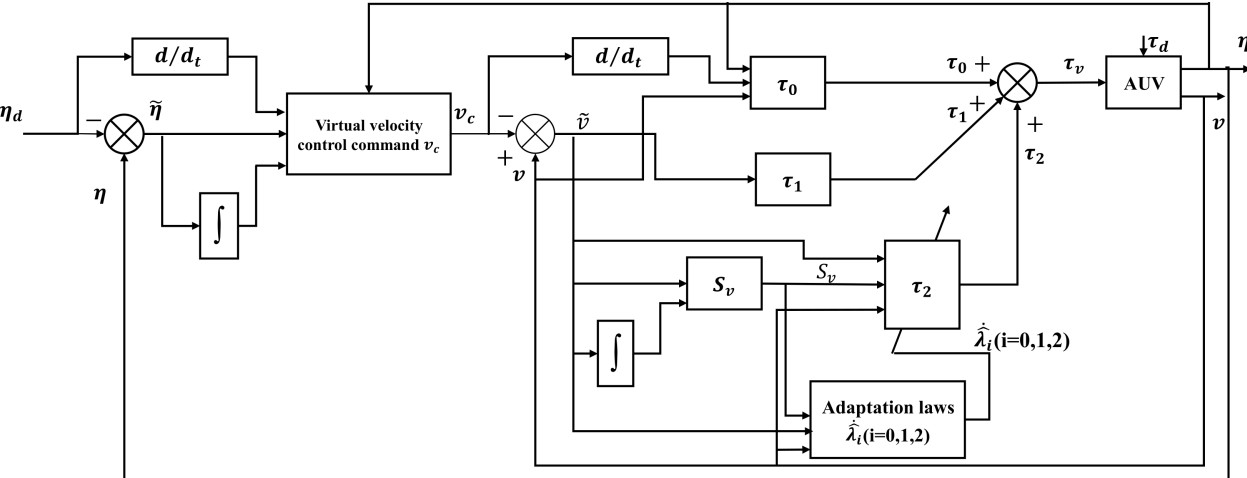

**Figure 5.** ANITSMC control block diagram (Reprinted from reference [91] with permission from the Institution of Engineering and Technology, copyright 2017).

Ref. [92] accomplished the finite-time error convergence and robust control task by designing a nonsingular fast fuzzy terminal sliding mode controller (NFFTSMC) with disturbance estimator for the six DOF dynamics of an AUV. The NFFTSMC controller expression with perturbation estimation is:

$$\boldsymbol{\tau_\eta} = \hat{\boldsymbol{M}}_\eta(\boldsymbol{\eta})(\ddot{\boldsymbol{\eta}}_d + \boldsymbol{\beta}\frac{q}{p}\boldsymbol{sig}^{2-\frac{q}{p}}(\dot{\boldsymbol{e}}) + \frac{1}{\alpha}\boldsymbol{\gamma}\boldsymbol{sig}^{\gamma-1}(\boldsymbol{e})\boldsymbol{\beta}\frac{q}{p}\boldsymbol{sig}^{2-\frac{q}{p}}(\dot{\boldsymbol{e}}) + \boldsymbol{k}_{fz}) - \hat{\boldsymbol{\tau}}_{dis} + \hat{\boldsymbol{N}}_\eta(\boldsymbol{v}, \boldsymbol{\eta}, \dot{\boldsymbol{\eta}}) \qquad (10)$$

where $\boldsymbol{\eta}$ denotes the vector of position and orientation of the vehicle expressed in inertial frame, $\boldsymbol{v}$ is the vector of linear and angular velocity expressed in body-fixed frame, $\boldsymbol{\alpha} = diag[\alpha_1, \cdots, \alpha_i, \cdots \alpha_n] \in \Re^{n \times n}$ in which $\alpha_i (i = 1, \cdots n) > 0$, $\boldsymbol{\beta}$ is a positive constant, $p$ and $q$ are positive odd integers satisfying $q > p$, $\hat{\boldsymbol{\tau}}_{dis}$ is the vector of estimated lumped uncertainty term, $\boldsymbol{\gamma}$ is an auxiliary vector to estimate the unknown perturbations from the dynamics of TSM manifold, $\hat{\boldsymbol{N}}_\eta(\boldsymbol{v}, \boldsymbol{\eta}, \dot{\boldsymbol{\eta}}) = \hat{\boldsymbol{C}}_\eta(\boldsymbol{v}, \boldsymbol{\eta})\dot{\boldsymbol{\eta}} + \hat{\boldsymbol{D}}_\eta(\boldsymbol{v}, \boldsymbol{\eta})\dot{\boldsymbol{\eta}} + \hat{\boldsymbol{G}}_\eta(\boldsymbol{\eta})$.

The SMC based control design suffers from the chattering phenomenon, which leads to a drop in the AUV control performance [93]. Ref. [94] employed an adaptive fuzzy sliding mode with PID sliding surface to deal with the depth trajectory tracking. Due to the integral term chattering problem not appearing. After that, ref. [95] used continuous adaptive PI terms instead of discontinuous switching items in the SMC. Adaptive fuzzy PI sliding mode control (AFPISMC) achieve high precise tracking ability in complex underwater environment. The method yields superior performance in terms of smooth and fast

trajectory tracking along with robustness against disturbances and perturbations. The AFPISMC control block diagram could be demonstrated in Figure 6.

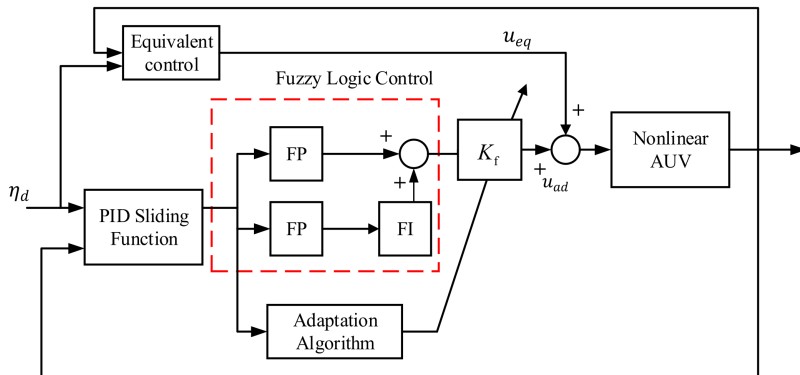

**Figure 6.** Block diagram of AFPISMC (Reprinted from reference [95] with permission from Springer-Verlag Berlin Heidelberg, copyright 2017).

Ref. [96] improved the sliding surface technique and obtained an adaptive nonfluctuating sliding mode controller with bounded estimates. The speed jump problem caused by the initial error is solved, the thrust of the propeller is avoided, and the control input and speed limit are satisfied. Considering the longitudinal speed control, we take the sliding surface $S_1$ as:

$$S_1 = u_e + \lambda_1 u_e + \lambda_2 \int u_e \tag{11}$$

Considering the heading angle and angular velocity tracking control, taking the sliding surface $S_2$ as:

$$S_2 = r_e + e_\psi + \lambda_3 \int (r_e + e_\psi) \tag{12}$$

In Equations (11) and (12), $S_1$ represents the first sliding surface, $S_2$ represents the second sliding surface, $u_e$ represents the longitudinal speed feedback control amount, $r_e$ represents the heading angle and angular velocity feedback control amount, $\lambda_1$, $\lambda_2$, $\lambda_3$ are positive constant, $e_\psi$ the heading angle deviation.

In addition, ref. [97] proposed a double closed-loop adaptive integral SMC without jitter. Replacing the sign function with a saturation function is used to overcome the jitter problem inherent in SMC. Double-loop chattering-free adaptive integral sliding mode control scheme is shown in Figure 7 as follows:

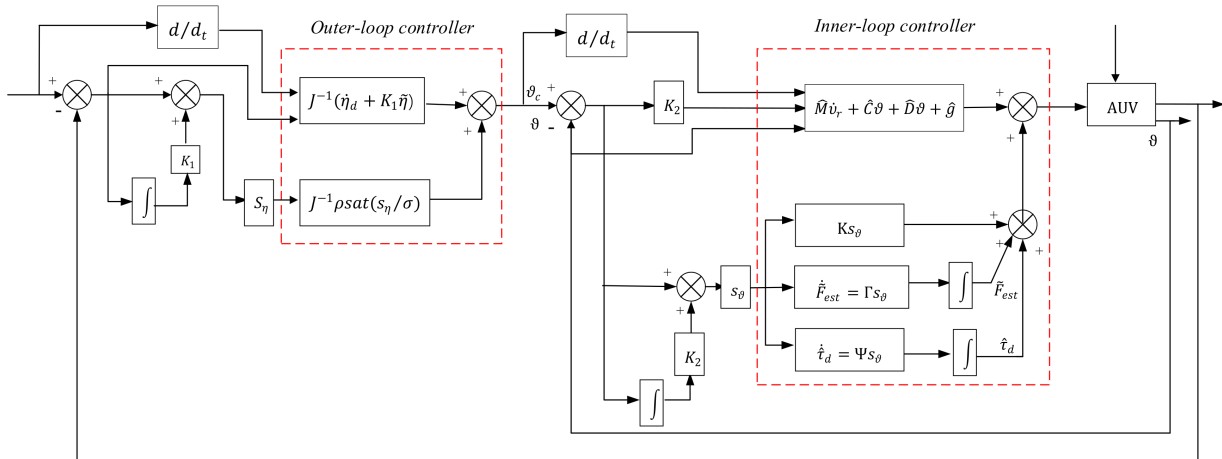

**Figure 7.** Block diagram for the double-loop chattering-free adaptive integral sliding mode control scheme (Reprinted from reference [97] with permission from Elsevier, copyright 2017).

Ref. [98] proposed a second-order SMC with a PID sliding surface, which is verified that the root mean square error of the second-order SMC with the switching controller is lower than that of the second-order sliding mode without the switching controller, and its closed-loop system is exponentially stable in the presence of parameter uncertainties and unknown disturbances. The second-order sliding surface is:

$$\dot{s}_i(t) + \beta_i s_i(t) = k_{pi}(t)e_i(t) + k_{ii}\int_0^t e_i(\tau)d_\tau + k_{di}\dot{e}(t) \tag{13}$$

where $k_{pi}, k_{ii}, k_{di}$ and $\beta_i$ are positive constants, for all $i$ = 1, 2, 3, $\beta_i$ determines the rate of decay for $s_i(t)$, the results show that the second-order sliding mode controller can compensate for the uncertainty of fluid dynamics and hydraulic parameters and eliminate external disturbances during movements.

In future research, for the inherent vibration phenomenon of the SMC system, filter or blur the SMC to make the output of the controller smoother. At the same time, a second-order SMC can be used to compensate for uncertain fluid dynamics and disturbances.

(2)    Backstepping control

Backstepping control is a nonlinear control that uses virtual control quantities that can handle environmental uncertainties [99]. It decomposes a nonlinear model into subsystems which do not exceed the system order. The design of intermediate virtual control quantities and Lyapunov functions for each subsystem has been investigated by using the backward recursion method [100]. By combining the linear stability theory, adaptive control, and observer design, the accurate tracking of the AUV trajectory in a complex environment is realized.

Considering that the external disturbance of the system dynamics model cannot be measured directly, ref. [101] used a high-gain observer to estimate system disturbances and compensate the control system. For the system dynamics of the longitudinal and lateral velocity bidirectional channels, the design of the bidirectional high gain observer is as follows:

$$\text{Longitudinal speed channel}: \begin{cases} p_1 = \hat{w}_1 - \mu_1 m_{11}u \\ \dot{p}_1 = -\mu_1(p_1 + \mu_1 m_{11}u) - \mu_1(m_{22}vr - d_{11}u + u_1) \end{cases} \tag{14}$$

$$\text{Transverse speed channel}: \begin{cases} p_2 = \hat{w}_2 - \mu_2 m_{22}v \\ \dot{p}_2 = -\mu_2(p_2 + \mu_2 m_{22}v) - \mu_2(-m_{11}ur - d_{22}v) \end{cases} \tag{15}$$

where $w_1, w_2$ is the external environment disturbance, $u, v, r$ are respectively the surge, sway of the AUV in the body coordinate system, $m_{11}, m_{22}$ is the inertial parameter including the added mass, $u_1$ is longitudinal thrust. By adjusting the size of the parameter $\mu_1, \mu_2$, the time constant of the observer could be changed to estimate the disturbance. $d_{11}, d_{22}$ is the AUV system hydrodynamic damping coefficient.

Based on the advantage of a backstepping technique, ref. [102] constructed the speed error function to obtain the appropriate control force and torque. This helps to improve the accuracy of trajectory tracking in 3D space. The second error equation is:

$$Z_2 = V - V_d = V_e \tag{16}$$

where $V$ is the surge sway, $V_d$ is the desired velocity vector, $V_e$ is the speed error.

However, the traditional backstepping control methods have a singularity problem. In response to this problem, ref. [103] used a backstepping design method in the design, and combined the input matrix decomposition of the high-frequency gain matrix. Although there are uncertainties in system parameters and interference forces, the submarine can still perform diving plane maneuvers. In addition, ref. [99] designed an adaptive rate for external interference, and used the backstepping method to define the virtual speed error variable. This control method can ensure that the underwater vehicle jump to the

designated area when there is external interference, and also solve the singular value problem in the traditional reverse thrust method.

Robust nonlinear controllers with a hierarchical structure (HRN) are designed for complex and simplified models of an AUV. Ref. [104] designed an HRN control technique that included a backstepping control for the first subsystem and a sliding-mode control for the second subsystem to achieve proper trajectory and orientation tracking.

Although several challenges as stated above have been addressed, external disturbance and system uncertainties are another two issues that need to be solved in tracking controller design for surface ships. To solve these problems, an observer-based estimation was preliminarily designed by [105] to estimate the disturbance and system uncertainties. Then, a backstepping controller was synthesized with a compensation control effort incorporated for accommodating the disturbance and uncertainties. The proposed controller guaranteed that the desired trajectory can be followed with an exponential rate of convergence.

The backstepping control is characterized by a complicated operation that is caused by the repeated difference of the virtual controller and increases computational burden of the control algorithm. As the system sequence increases, so does the complexity of the controller. Dynamic surface control techniques could be used to solve this problem in future research. The comparison between different nonlinear time invariant control methods in AUV trajectory tracking is shown in Table 4.

**Table 4.** Nonlinear time invariant control algorithm in AUV trajectory tracking.

| Control Algorithm | Research Purposes | Improvement | Control Effect | Ref. |
|---|---|---|---|---|
| **Sliding mode control** | Improve control accuracy | Robust sliding mode controller | Successfully controls the AUV roll angle, pitch angle and yaw angle within 10 s. | [106] |
| | | Sliding mode variable structure control | The AUV can accurately reach the termination point from the starting point. | [107] |
| | | Fuzzy logic | The trajectory tracking accuracy of the vehicle in all directions is very high. | [46] |
| | Solve the jitter problem | Bounded adaptive estimation | Solve the problem of speed jump due to initial error in conventional backstepping method, avoiding thruster saturation and satisfying control input and speed constraint conditions. | [96] |
| | | Dual closed-loop adaptive integral sliding mode controller | The designed controller can effectively eliminate the flutter effect. | [97] |
| | | Self-adaptive fuzzy PI sliding mode control | PISMC has less oscillator response and the shortest delay time. | [95] |
| | | Adaptive fuzzy sliding mode with PID sliding surface | Avoid response oscillating and reduce arrival time. | [94] |
| | Achieve the finite-time convergence of the system dynamics | The terminal sliding mode | Force the AUV's position to track the desired time-varying trajectory. | [54] |
| | | Adaptive nonsingular integral terminal sliding mode control | Better robustness and faster convergence. | [91] |

**Table 4.** *Cont.*

| Control Algorithm | Research Purposes | Improvement | Control Effect | Ref. |
|---|---|---|---|---|
| **Sliding mode control** | Solve the limitations of actuators | The second-order sliding mode controlled | Effectively compensate for the uncertainties of the hydrodynamic and hydrostatic parameters of the vehicle and can eliminate unpredictable disturbance effects. | [108] |
| | | A second-order sliding mode controller using PID sliding surface | 2-SMC with switching controller showed smaller rms error in steady state than 2-SMC without switching controller. | [98] |
| | Realize horizontal trajectory tracking | Line-of-sight method | The underwater vehicle can be accurately set according to the preset, except for deviations at the starting point and the turning point. | [109] |
| | | Combination of the lateral trajectory error method and the line-of-sight method | Guarantees global κ-exponential stability of the cross-track error to straight line trajectories in three-dimensional space. | [110] |
| | | Combining the cross-tracking error method and the line-of-sight method | The sliding mode controller has good tracking performance for time-varying depth signals. | [111] |
| **Backstepping control** | Estimate faster convergence of parameters and tracking errors | Adaptive control scheme | Realize three-dimensional track precise tracking control. | [104, 112] |
| | AUV's virtual speed control and trajectory tracking enable asymptotic stability | Hierarchical control | The robustness of the system under environmental disturbance is guaranteed. | [105, 113] |

Adaptive Control

The central concept of adaptive control is to estimate unknown parameters online using known system conditions. It suppresses the influence of external disturbances, environmental changes, and the effect of the system coupling with itself. Adaptive controls also decrease modeling errors and their influences [55].

Ref. [114] developed an adaptive control law for AUV to track the desired trajectory. The desired state-based regression matrix controller provides consistent results under fluid dynamic parameter uncertainties. The proposed adaptation control law is shown in Figure 8.

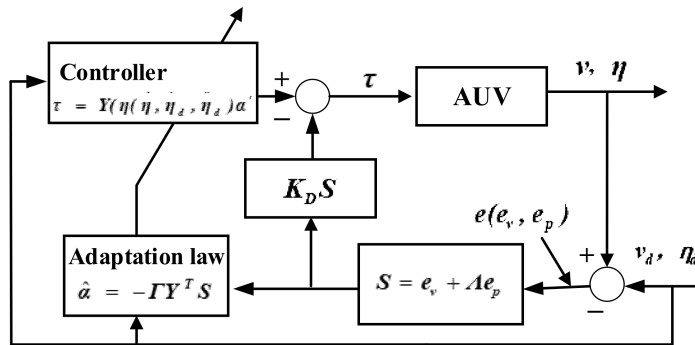

**Figure 8.** Structure of the adaptation control law.

Based on the attractive manifold design approach, ref. [115] developed an adaptive autopilot with equivalent uncertainty for diving level control of submersibles. The results show that in the closed-loop system with σ-modification, the pitch angle and depth tracking error are finally uniformly limited.

In addition, ref. [32] proposed an improved sight-based adaptive controller for underwater vehicles. The terminal SMC is used to improve the robustness and asymptotic convergence, and a wind-resistant compensator is used to reduce the effect of actuator saturation.

Under the premise of ensuring the accuracy of adaptive control, a fast convergence parameter estimation algorithm should be studied. In addition, combining adaptive control with other control methods (such as backstepping control, SMC, and NN control) will result in better control schemes.

Robust Control

Robust control describes a system that has parameter uncertainties and limits unmodeled dynamics [58]. Different from adaptive control, robust control needs to maintain certain performance indicators under a known control structure [23].

At present, two robust controllers are used in AUV trajectory tracking control research: (i) H∞ control, and (ii) exponential convergence robust control. Ref. [116] designed the heading controller and the trim controller based on robust H∞ control theory. This effectively realized 3D AUV tracking control and steady state control accuracy. For an AUV trajectory tracking control with dynamic uncertainties and time-varying external disturbances, ref. [23] proposed three index-stable controllers: (i) the minimum-maximum controller, (ii) the saturation controller, and (iii) the smooth transition controller. The results showed that the filter, position, and seed tracking errors of the three controllers had exponential convergence characteristics. Block diagram of the three proposed exponentially convergent robust controllers is shown as Figure 9.

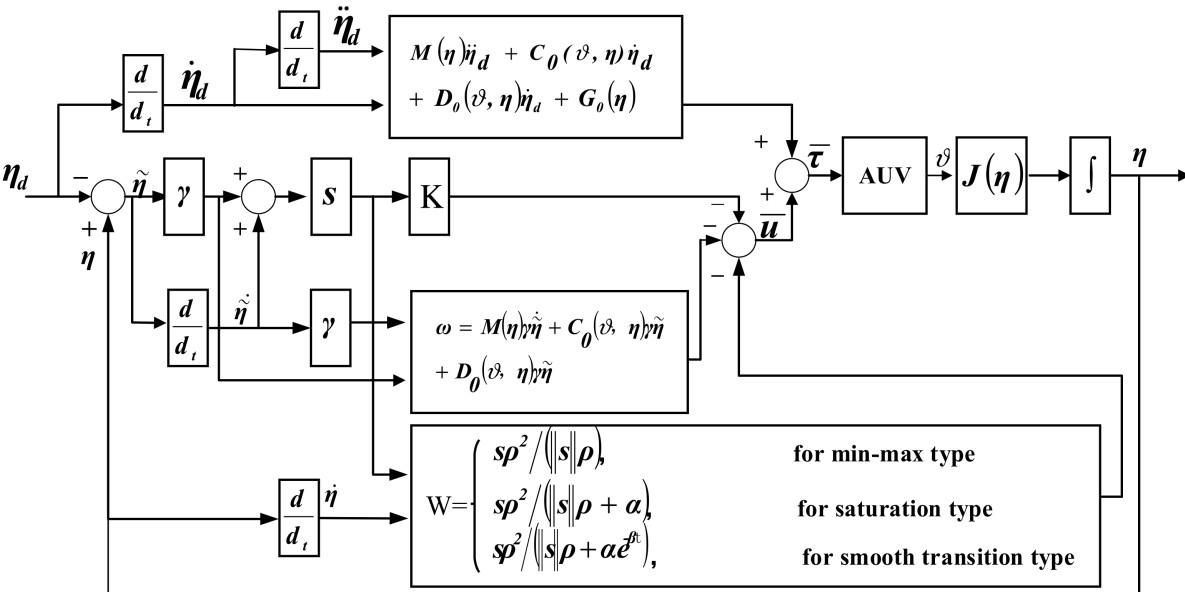

**Figure 9.** Block diagram of the three proposed exponentially convergent robust controllers (Reprinted from reference [23] with permission from Elsevier, copyright 2017).

Intelligent Control

(1)     Fuzzy control

The fuzzy controller is based on the manual control strategy of the operator or on fuzzy information that the designer knows about the process [117]. The fuzzy control method has been proposed for AUV trajectory tracking.

Ref. [118] created a virtue of the adaptive fuzzy-based dynamic surface control scheme. The tracking errors using this scheme can converge to zero faster and without chattering. To simultaneously control the speed and attitude of the AUV, ref. [119] proposed a self-tuning nonlinear fuzzy PID controller. The output signal from the fuzzy control is used to fine tune the PID parameters. The block diagram of self-tuning nonlinear fuzzy PID controller is shown in Figure 10.

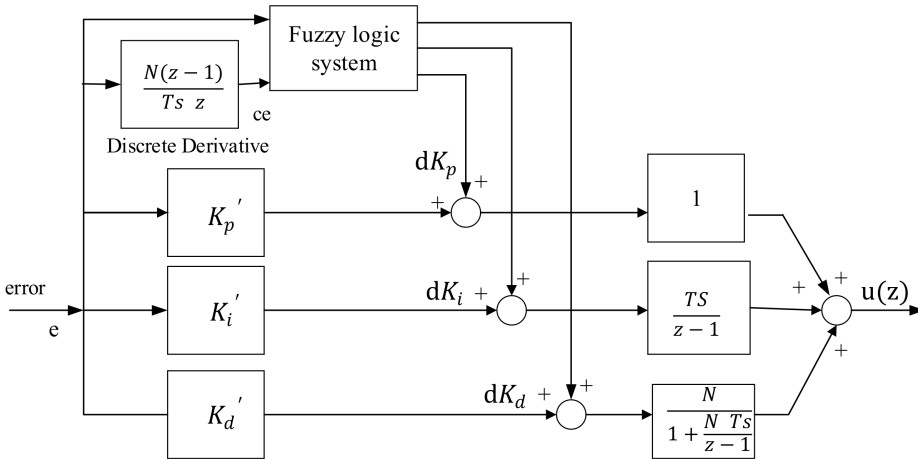

**Figure 10.** Block diagram of self-tuning nonlinear fuzzy PID controller (Reprinted from reference [119] with permission from Hammad, copyright 2017).

In addition, ref. [120] used layered closed-loop fuzzy control to achieve underactuated AUV horizontal trajectory tracking control. The fuzzy inference system is used to achieve the highest level of control to obtain the guidance controller. However, the above studies did not consider the actuation saturation of the vehicle, which means that the control input is not limited. Ref. [121] controlled the vertical and horizontal planes of underactuated AUV with actuator saturation and unknown disturbances. Direct adaptive fuzzy control is used to compensate the effect of actuator saturation, which ensures the stability of the trajectory tracking system when the actuator is saturated.

The aforementioned research shows that the combination of fuzzy control and other control methods can achieve better trajectory tracking effects. In addition, the traditional membership function is determined based on the researcher's experience. Therefore, it is essential to further develop turning and learning techniques.

(2) Neural network (NN) control

NN is a collection of neurons or nodes with adjustable connection weights [122]. Due to the NN's ability to process data, learn nonlinear systems [123], and provide approximations [124], it has achieved good results in AUV trajectory tracking.

The NN's approximation ability and adaptive methodology can compensate for unknown parameters like coupling and external disturbances of the system [73]. Ref. [125] combined the unscented KF pose estimator with an adaptive NN tracking method to estimate the AUV altitude. Ref. [126] studied the adaptive robust control problem based on NN. The results indicated that it is possible to adjust only one parameter without determining the number of NN nodes. Ref. [37] used the DSC method to design the AUV trajectory tracking control model and the NN to estimate AUV model uncertainties.

Linear parameterized radial basis functions (RBFs) are widely used as adaptive functions in NN-based adaptive controllers to simulate nonlinear dynamics and all object uncertainties [122]. To overcome parameter uncertainties and realize the trajectory tracking, ref. [127] introduced the RBF NN to estimate unknown parameters and select an adaptive law to guarantee the best estimate of NN weights. Ref. [123] combined a neural adaptive controller and an RBF NN with passive boundary conditions, feedback linearization, and

approximation ability of AUV swing and echo velocity. Simulation results show that the AUV effectively tracked the required trajectory through smooth transient performance.

Due to changes in the hydrodynamic coefficients, parameter uncertainties and modeling errors can result in poor controller performance. Ref. [128] used NNs to enhance the structure of the dynamic linear compensator. The NN extended the working range of the vehicle and achieved control of the nonlinear system. Ref. [124] conducted AUV trajectory tracking control research in discrete time with a continuous AUV model. NN reinforcement learning was used to overcome the effects of model parameter uncertainty and environmental disturbances. Two NNs were used for controller design. One NN was used to compensate for controller uncertainty and the other NN was used to estimate the evaluation function for optimal AUV tracking performance.

Although NNs have been successfully used in AUV trajectory tracking control, it entails a large amount of calculation. Therefore, it is necessary to reduce neurons and simplify the network. At the same time, adopting a better network structure and learning scheme could solve inherent problems such as a local minimum. Recent work in this direction includes deep learning and extreme learning. However, relatively few results have been applied to the control field.

(3) Reinforcement learning (RL)

Current developments in the field of deep RL have made neural networks viable approximators of the value and policy function [129]. The progress was expanded by the development of a number of policy function and actor–critic algorithms which allow the RL agent to select continuous values for the control actions in order to solve complex continuous control tasks [130]. In the RL control scheme, the input parameters are the data that can be measured by the on-board sensors directly, and the outputs of the designed controller are set to the actions of the vectored thruster [131].

Policy-gradient-based models are particularly sensitive to the shape of the reward function. The smallest variations can lead to the convergence or not of the model. Therefore, a trade-off between the reward function and the complexity of the environment must be found. In order to perform a waypoint tracking mission, ref. [132] designed the reward function, where the reward function takes into account low-level variables such as linear and angular velocities and their respective references. The result show that the thrusters were 11.14% less solicited by the latter controller. Considering different factors which actually affect the control accuracy of AUV navigation control, ref. [133] developed a reward function for deep RL controller. The designed reward function can effectively improve reliability and stability, reduce energy consumption, and restrain the vectored thruster sudden change. However, these methods are still difficult to apply directly to the actual AUV system because of the sparse rewards and low learning efficiency. Ref. [134] proposed a deep interactive RL method for path following of AUV by combining the advantages of deep reinforcement learning and interactive RL. They further propose a deep RL method that learns from both human rewards and environmental rewards at the same time. The result show that, AUV can converge faster than a DQN learner from only environmental reward.

Model-free predictive control is a novel data-driven control approach. It can calculate directly the control input by using a great deal of input and output datasets. Ref. [135] proposes a model-free goal-driven deep RL method, based on the DDPG algorithm, for self-tuning of the low-level PID controllers of mobile robots. The formulation includes a gradient inverting scheme for constraining the actor outputs along the training phase. The introduction of universal value functions, and its extension to the policy network, allows for improved adaptability, making the agent able to adapt to different operative conditions.

When RL is applied to AUV trajectory tracking systems, the state and action variables are continuous, and due to the generality of the dynamics and reward functions considered, it is usually impossible to derive exact, closed-form representations of the value function or control policy. In the context of AUV trajectory tracking control, more research is needed into determining which kinds of problems benefit from using DNNs as function

approximators. This might have to be done by getting a better understanding of how the representations are learned for common network types and what types of functions are represented and learned efficiently. The comparison between different intelligent control methods in AUV trajectory tracking is shown in Table 5.

**Table 5.** Intelligent control algorithm in AUV trajectory tracking.

| Control Algorithm | Classification | Improvement | Control Object | Control Effect | Ref. |
|---|---|---|---|---|---|
| **Fuzzy control** | Fuzzy PID | Self-tuning nonlinear fuzzy PID controller | Control position and speed to follow desired trajectories. | Compared with traditional PID, the response speed is faster and the minimum error time is reduced. | [119] |
| | | Hierarchical closed-loop fuzzy control | Closed loop planar trajectory tracking. | Motion and velocity errors are bounded and fast converging, showing the robustness of the control algorithm for external disturbances. | [121] |
| | | Direct adaptive control | Compensate for the effect of actuator saturation. | System stability for trajectory tracking in the presence of actuator saturation. | [20] |
| **Neural network control** | Adaptive neural network | Unscented Kalman filter | Pose estimation. | Ensure the accuracy and certainty of the estimate, as well as the feasibility of trajectory tracking control. | [125] |
| | | Filtered technique | Trajectory tracking of AUV with model errors and external disturbances. | Avoided "explosion of complexity". | [126] |
| | | Linearly parameterized radial basis function | Estimate unknown terms. | Removing the inherent error. | [123, 127] |
| | | Nonlinear adaptive controller | Precise trajectory tracking. | A satisfactory approximation capacity and clearly result in superior tracking performance. | [74] |
| | Online neural network controller | Dynamic linear compensator | Compensating model error. | Extend the operating range of the AUV beyond the capacity of the linear controller. | [128] |
| | | Reinforcement learning | Address unknown disturbances, parameter uncertainties and control input nonlinearities. | Obtain the optimal tracking performance. | [124, 136] |
| | Hybrid control | Dynamic surface control | Tracking curve or straight line. | Reduces controller complexity. | [37,47] |

**Table 5.** *Cont.*

| Control Algorithm | Classification | Improvement | Control Object | Control Effect | Ref. |
|---|---|---|---|---|---|
| **Reinforcement learning** | Reinforcement learning | Designed the reward function | Precise trajectory tracking. | The thrusters were 11.14% less solicited by the latter controller. | [132] |
| | Deep reinforcement learning | A reward function for deep RL | Improve AUV trajectory tracking precise. | Effectively improve reliability and stability, reduce energy consumption, and restrain the vectored thruster sudden change. | [133] |
| | Interactive reinforcement learning | Learns from both human rewards and environmental rewards at the same time | Improve rewards and learning efficiency. | AUV can converge faster than a DQN learner from only environmental reward. | [134] |
| | Model-free goal-driven deep RL | Based on the DDPG algorithm | Self-tuning of the low-level PID controllers of mobile robots. | Improved adaptability, making the agent able to adapt to different operative conditions | [135] |

Others

(1) Cascade systems

A nonlinear cascade system is composed of two subsystems in a cascade structure [137]. The control enters only one of the subsystems, and the change in the state of another subsystem is achieved through the association between the two subsystems. This design method is characterized by simplified controller design, ignoring some nonlinear terms. This results in uncomplicated expression control laws.

Researchers have used cascade systems to analyze and control the horizontal plane [138] and three-dimensional movement [139] of AUVs. To improve water level tracking control of AUVs, ref. [138] proposed a control torque design based on the cascade system theory. The system is broken down into two cascade systems: position tracking and heading angle tracking. The inversion method obtains a globally consistent, asymptotically stable linear track following controller for the heading angle tracking system. Ref. [139] studied the global trajectory tracking control. Based on the results of the time-varying cascade system, the tracking error kinematics and dynamics were divided into two separate subsystems. Ref. [67] used the nonlinear cascade system stability theory to decompose the 3D linear tracking system model into a cascade of two horizontal systems (horizontal tracking and vertical plane linear tracking), and then select the appropriate altitude angle. The instructions are further broken down into cascade position tracking and altitude tracking systems.

(2) Bio-inspired control

The bio-inspired dynamics model has the ability to smooth and bound the error output, thereby obtaining smooth, physics-constrained speeds, forces, and moments [140]. This method takes into account the constraints of mechanical dynamics and successfully solves the problem of dynamic constraint matching and jumping speed of underwater vehicles [141].

Ref. [142] combined a bio-inspired NN model with a Lyapunov function, which solved the problem of speed jumps in discrete paths. Ref. [143] solved the problems of impact and sharp jumps in angular velocity that cannot be achieved during the initial movement of the AUV through the biologically inspired shunt model and nonlinear feedback trajectory tracking control method. Ref. [144] proposed a 3D motion control based on a bio-inspired neural dynamics model by constructing a simple intermediate dummy

variable and combining the Lyapunov function to design trajectory tracking control law. This achieves global asymptotic stability and smooth continuous output of horizontal and vertical planes.

Based on a knowledge base, ref. [145] presented an original ship course-keeping algorithm. Its integral part is a computer-borne ship movement dynamical model based on a set of signals obtained from the object's input and output, which has been avoided the problems occurring while designing classic control algorithms for a complex, non-linear ship model. The designed algorithm was compared to LQR controller as well as feedback linearizing one. The results prove high quality performance of the proposed method.

### 3.2.2. Physical Experimental Study

In order to verify the reliability of the mathematical model and numerical simulation of the control strategy, it is more important to conduct physical experiment research in AUV. In order to evaluate the control method proposed in AUV trajectory tracking, the control algorithm is compiled into C/C++ language, MATLAB, etc., and then integrated into the control software. At the same time, sensors are installed in the underwater robot to measure its depth, speed, and other data. However, there are still errors between the actual experimental data and the simulation results [146]. This article introduces some typical experimental cases.

In order to prove the feasibility of the developed controller, ref. [98] adopted the AUV Cyclops, which is equipped with a Doppler velocity log to obtain the positions and velocities of AUV, a fiber-optic gyro unit to obtain heading and yaw rate, and a digital pressure transducer to obtain the AUV's depth. They applied 2-SMC with the switching controller, which is verified that the root mean square error of the second-order SMC with the switching controller is lower than that of the second-order sliding mode without the switching controller.

Ref. [147] translated the control strategies into C/C++ programming, and integrates it into the vehicle control software. A depth sensor with an accuracy of 0.003 m is installed on the UVIC-I AUV, and the heave speed is obtained through the difference of the depth sensor data. An MPC whose coefficient changes with the error is introduced to adjust the control increment vector weighting matrix. The results show that when the tracking step length is 1 m, this method can reduce the setup time by about 2 s.

To verify the tracking performance and robustness of the proposed scheme, ref. [148] used a small remote-controlled underwater vehicle to carry out an experimental procedure in a water tank. The deployed vehicle, VideoRay PRO, is equipped with three thrusters that affect the surge yaw motion. In addition, the Polhemus-Isotrack device connected to the host computer via 30 Hz RS-232 serial communication is used as the attitude feedback sensor for the motion control scheme. As predicted by the theoretical analysis, despite the lack of knowledge about the parameters of the vehicle dynamics model, the tracking with the specified performance is successfully achieved, and the error is strictly evolving within the predefined performance range.

It can be seen from literatures, AUVs are usually equipped with sensors, which can obtain the position and speed of the AUV. Therefore, the data-driven control methods have great significance to AUV trajectory tracking control in a complex environment. Deep learning, reinforcement learning, and model-free predictive control methods will be of great significance in AUV trajectory tracking control.

### 3.3. Control Performance

The performance of AUV trajectory tracking determines its operation accuracy. The main control performance indicators include control accuracy, system response speed, trajectory tracking convergence speed, stability, and robustness. Table 6 shows the performance comparison of different controllers in AUV trajectory tracking. From the above literature, we can see that the following contents are worth noting: [39] developed a disturbance observer-based nonlinear MPC scheme for cross tracking of underactuated AUV

under sea current disturbances. The stability time is 110 s, and the error is under the 5% limit. Ref. [53] used an SDRE controller in a non-affine structure without decoupling the six DOF. The position error of 43.6 mm is negligible compared with the total measured distance of 8.77 m. For depth trajectory tracking control, ref. [147] introduced a MPC whose coefficient changes with the error to adjust the control increment vector weighting matrix. The results show that when the tracking step length is 1 m, the method can reduce the setup time by about 2 s.

For AUV dynamics, time-varying, nonlinear, and unpredictable external environmental uncertainties, the control system must be adaptive and robust, and the classic controller such as PID controller has a fixed gain. When the state of the vehicle and the environment change significantly, there is no guarantee that the entire system will respond well. Modern control theory is much broader than classical control theory can handle, including linear and nonlinear systems, stationary systems and time-varying systems, and univariate systems and multivariable systems. Although the above controller has achieved good control effects in AUV trajectory tracking control, it still faces the following challenges: (i) Fails to consider the achievability of control speed and mechanical dynamic constraints, such as the actual AUV acceleration range. (ii) AUVs may suffer from failures in unknown and complex marine environments, and they require a stable and reliable control method. (iii) Uncertainties caused by model errors and unmodeled dynamics are often expressed as unknown nonlinear functions of system states, control inputs, etc., and may also contain unknown parameters, making system controller design particularly difficult. Intelligent control is not based on an accurate mathematical model of the controlled object. It is based on knowledge, based on input/output data/information causality, and intelligent inference control theory. Intelligent methods are therefore used to solve control problems in complex systems, such as fuzzy control, neural network control, and reinforcement learning.

**Table 6.** Summary of different type of controller in trajectory tracking control.

| Control Types | Strength | Weakness | Future Improvement | Whether Based on Model |
|---|---|---|---|---|
| PID | Flexibility, simplicity, and good performance. | Poor resistance to external interference. | Developed by combining other control algorithms. | Y |
| LQR | Can set the unstable system, and the method is simple and easy to implement. | Lacks the characteristics of robustness. | Online iterative learning linear quadratic regulator (OILLQR). | Y |
| SDRE | Ensure a wide range of progressive stability | Only applicable to the affine nonlinear system in the form of state correlation coefficient (SDC). Can only guarantee the local asymptotic stability of the closed-loop system. | Each mathematical model of each cycle can be processed into a form similar to a linear system, and the feedforward method is used to compensate for the nonlinear redundancy terms. | Y |
| MPC | Effectively overcome the uncertainty of controlled objects, the dynamic effects of lag and time-varying factors. | Heavy online computational burden. | Offline precomputation, delay compensation, event triggering strategies, and digital continuations. | Y |
| SMC | Has a certain resistance to modeling errors, time-varying parameters, and external environment interference. | Jitter problem. | The filtering or fuzzy sliding mode control method. | Y |

**Table 6.** *Cont.*

| Control Types | Strength | Weakness | Future Improvement | Whether Based on Model |
|---|---|---|---|---|
| **Backstepping control** | The actuator's control output is continuous and the control system does not experience jitter. | Vehicle tracking speed jump problem.Inherent disadvantage of "explosion of complexity". | The "dynamic surface control" technique. | N |
| **Adaptive control** | Ability to re-adjust controller parameters online | Asymptotic convergence under ideal conditions of time infinity. | Combined with other control methods. | N |
| **Robustness control** | Ensure a certain level of dynamic performance while maintaining stability. | Cannot counter the complex control system in actual engineering | The combination of different methods can make the control scheme more effective. | Y |
| **Fuzzy Control** | The control system can maintain good performance even when the characteristics of the controlled object change or perturb. | When establishing methods of fuzzification and inverse fuzzification, there is a lack of systematic methods. | The neural network can dynamically adjust the membership function and the fuzzy rule according to the system information. | N |
| **NN** | Greater degree of fault tolerance and strong data processing capabilities. | Exists a limitation referred to as "the curse of dimensionality". | Use the "Minimum Learning Parameter (MLP)" algorithm to reduce the computational burden of the algorithm. | N |
| **Cascaded system** | Simplifying the controller design, the expression control law is not complicated. | – | – | N |
| **Bio-inspired** | Eliminate the speed jump problem. | – | – | Y |

## 4. Conclusions and Future Perspectives

Through the analysis of the existing technology, it is found that the modeling method and control strategy play a key role in high-precision and fast trajectory tracking, which also affects the accuracy of underwater operations. Through advanced modeling methods or control strategies, more attention has been paid to the accuracy of trajectory tracking. There is a conflict between the error convergence speed and the high-precision control system. Therefore, balancing the relationship between the two is a challenge. In this work, the latest literature on the methodology of modeling methods and control strategies aimed at improving the error convergence speed and high precision of AUV trajectory tracking has been systematically studied.

The review has been developed around a proposed classification for modeling approaches and another classification for control strategies. Moreover, a discussion with the aim of establishing suitable modeling, control approaches, and research gaps that need to be addressed is also established. In this way, from the discussion, it is concluded that most of the simplified and approximated models reported are an oversimplification of the AUV trajectory tracking systems, which are not useful to test the behavior of the controllers under realistic conditions, and the system identification models are an attractive option for control systems design and testing. Considering the powerful capabilities of deep learning and reinforcement learning in data processing, these methods will have great potential in the establishment of AUV trajectory tracking models.

Furthermore, the review presents three case studies, which illustrate the development of a control-oriented modeling strategy, and the design of the most common control approaches. In the case study, three control strategies are explained and implemented,

showing their procedure, specific results, and features. At present, most of the articles in AUV trajectory tracking control use MATLAB/Simulink simulation research. However, there are few actual underwater experimental studies that can verify the unreliability of mathematical models and numerical simulations. At the same time, when designing the controller, efficiency is also an important criterion to be considered, because it could achieve the control target with less energy consumption in a limited time, thereby obtaining better performance.

In addition, the control performance with different control strategies have been compared. In order to meet the requirements of control accuracy, many studies have proposed an advanced controller combined with artificial NN, which can fully meet the requirements of control accuracy. In addition, in order to minimize tracking errors, artificial NN seems to tend to increase convergence speed and achieve high accuracy, especially when combined with traditional control algorithms (such as PID, fuzzy control, MPC) it achieved great performance in AUV trajectories tracking.

Finally, the performance of AUV trajectory tracking control is of great significance for exploring the underwater environment and performing underwater operation tasks. The theoretical contribution of trajectory tracking control also helps to improve operation performance. According to the review and the results obtained, the development of new control-oriented models, the research in the estimation of unknown inputs, and the development of more innovative control strategies for AUV trajectory tracking systems are still open problems that must be addressed in the short term. At the same time, the combination of environmental data-driven methods, artificial simulation (machine learning for human-computer interaction applications), and powerful fault-tolerant systems with AUV control will have far-reaching significance.

**Author Contributions:** Writing original draft preparation, L.D.; writing—review and editing, D.L. Both authors have read and agreed to the published version of the manuscript.

**Funding:** This research was funded by National Key R&D Program of China "Next generation precision aquaculture: R&D on intelligent measurement, control and equipment technologies" [2017YFE0122100]; Research on the rapid detection mechanism and method of trace-level toxic nitrogen in aquaculture water based on SERS optopole [2018QC188].

**Institutional Review Board Statement:** Not applicable.

**Informed Consent Statement:** Not applicable.

**Data Availability Statement:** Not applicable.

**Conflicts of Interest:** The funders had no role in the design of the study; in the collection, analyses, or interpretation of data; in the writing of the manuscript, or in the decision to publish the results.

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
