# Peer review of "AUV Trajectory Tracking Models and Control Strategies: A Review"

_jmse, doi:10.3390/jmse9091020_

Round 1

Reviewer 1 Report

This review paper provide a comprehensive survey and reviews on the current control models and their parameters, algorithms and strategies. Also this article contributes to identify the future development direction for AUV research. 

Comments:

  1. In the section 2.1 Analytical model, many equations and figures are referred from the book 'Handbook of marine craft hydrodynamics and motion control' written by Fossen, T.I., and it doesn't seem to be necessary to include these contents in this paper considering the length of the paper.
  2. The article fails to contribute novel technical insights related to the novel control methods for the AUV trajectory tracking design and any real case study or experiment. Also, no technical comparison of the proposed method with others. 

Author Response

Dear Editor and reviewer,

Thank you very much for your insightful guidance and constructive comments! We have revised our manuscript according to your instructions. Your comments have led to our further improvement of the paper. The revisions are highlighted in yellow in the revised manuscript. Point-by-point responses to your comments are given.

Sincerely yours,

Daoliang Li

College of Information and Electrical Engineering

China Agricultural University

General Comments: This review paper provides a comprehensive survey and reviews on the current control models and their parameters, algorithms and strategies. Also, this article contributes to identify the future development direction for AUV research.

Response: Thank you very much for your positive comments and approvals. We have repeatedly read and revised this paper following your valuable advice. The specific revisions can be seen in the revised paper, where all the revisions are highlighted in yellow.

Point 1: In the section 2.1 Analytical model, many equations and figures are referred from the book 'Handbook of marine craft hydrodynamics and motion control' written by Fossen, T.I., and it doesn't seem to be necessary to include these contents in this paper considering the length of the paper.

Response 1: Thanks for your constructive comments. Following your comments, we deleted equations (1)-(7), and removed the explanation of the equation's parameters. At the same time, considering the length of the paper, we deleted Figure 3. In addition, we have changed the sequence numbers of equations and figures in the full text.

Point 2: The article fails to contribute novel technical insights related to the novel control methods for the AUV trajectory tracking design and any real case study or experiment. Also, no technical comparison of the proposed method with others.

Response 2: Thanks for your constructive comments. Following your comments, we have read a lot of the latest literatures, and found that over the last years, a significant amount of papers appeared devoted to the application of reinforcement learning methods. We found these studies highly interesting and important since even being model-free in some cases, they are able to compete with model-based classic methods. In order to provide novel control methods for AUV trajectory tracking design and real case study or experiment, we added a section reinforcement learning in the paper. We introduced the principle of reinforcement learning and its application in detail, and proposed challenges and solutions for AUV trajectory tracking control. The specific modifications please see line 666-705. And also, through the summary and analysis of existing methods, we propose that the development of new control-oriented models, the research in the estimation of unknown inputs, and the development of more innovative control strategies for AUV trajectory tracking systems are still open problems that must be addressed in the short term. At the same time, the combination of environmental data-driven methods, artificial simulation (machine learning for human-computer interaction applications), and powerful fault-tolerant systems with AUV control will have far-reaching significance. The specific modifications please see line 858-867.

In section 3.2.2, we analysed the typical experiment case in AUV trajectory tracking. We introduced the programming software in each experiment, hardware system of AUV, and the control strategies. The results in the experiment are discussed. It can be seen from literatures, AUVs are usually equipped with sensors, which can obtain the position and speed of the AUV. Therefore, the data-driven control methods have great significance to AUV trajectory tracking control in a complex environment. Deep learning, reinforcement learning and model-free predictive control methods will be of great significance in AUV trajectory tracking control. The specific modifications please see line 781-785.

We revised the article and compared the proposed methods in detail. Table 2 compared the existing analytical modeling and system identification methods. And in the article, we compared the advantages and disadvantages of analysis modeling and system identification in detail, and proposed the multi-model framework realizes modeling and identification of complex nonlinear systems through problem decomposition. Please see line 295-302. In addition, we added table3-table5. We compared the specific methods of AUV trajectory tracking control in each category, including method improvements, control targets, and control effects. And in table 6, we compared the control strategies proposed by AUV trajectory tracking control, including the advantages and disadvantages, future improvement methods, and whether based on model control. We hope to provide guidance for researchers to choose appropriate control strategies in AUV trajectory tracking control. Moreover, we added a discussion about the control strategies, and point out that due to the huge challenges in the AUV movement process, the intelligent control method has great potential in AUV trajectory tracking, due to it is not rely on mathematical models, but is based on knowledge, based on input/output data/information causality. Please see line 800-818.

We have colored the related changes in yellow in the revised manuscript for your convenience. Please view the changes in revised manuscript. We would like to thank you again for your valuable comments that truly help us improve this paper for the final publication.

Sincerely,

Daoliang Li

Ling Du

Reviewer 2 Report

The paper is related to AUV trajectory tracking models and control strategies. The topic is interesting to the academic community. In order to enhance the review quality, I suggest the following remarks be taken into account:

  1. Please highlight the elements of the novelty.
  2. The authors are encouraged to indicate their contribution to the development of AUV trajectory tracking models and control strategies.
  3. Figure 1 seems to be insufficiently described.
  4. Please mark the vectors in bold.
  5. The References should be extended to include the publications on intelligent solutions in shipping that refer to the discussed subject, for instance:
    • Qu, Y.; Xiao, B.; Fu, Z.; Yuan, D. Trajectory exponential tracking control of unmanned surface ships with external disturbance and system uncertainties. ISA Transactions 2018, 78, 47-55.
    • Borkowski P., Zwierzewicz Z. „Ship course-keeping algorithm based on knowledge base” Intelligent Automation and Soft Computing vol. 17, no. 2, 2011 (149-163)

Author Response

Dear Editor and reviewers,

Thank you very much for your insightful guidance and constructive comments! We have revised our manuscript according to your instructions. Your comments have led to our further improvement of the paper. The revisions are highlighted in yellow in the revised manuscript. Point-by-point responses to your comments are given.

Sincerely yours,

Daoliang Li

College of Information and Electrical Engineering

China Agricultural University

General Comments:  The paper is related to AUV trajectory tracking models and control strategies. The topic is interesting to the academic community. In order to enhance the review quality, I suggest the following remarks be taken into account:

Response: Thank you very much for your positive comments and approvals. We have repeatedly read and revised this paper following your valuable advice. The specific revisions can be seen in the revised paper, where all the revisions are highlighted in yellow.

Point 1: Please highlight the elements of the novelty.

Response 1: Thanks for your constructive comments. The elements of the novelty lie in the following aspects: First, we overviewed AUV trajectory tracking modeling technologies, compared the analytical models and system identification-based models. We pointed out that the multi-model framework could realize modeling and identification of complex nonlinear systems through problem decomposition. Second, we analysed the challenges in AUV trajectory tracking control, overviewed AUV trajectory tracking control strategies. We conducted discussions on the existing control methods, and put forward prospects for each method in the AUV trajectory tracking control. Third, we discussed potential topics for future research in the field. Fourth, we propose the environmental-data driven methodologies have great significance in future research.

In addition to summarizing the literature, this article has conducted a lot of discussions on the existing methods and technologies in the article, expounded a lot of my own views, and put forward prospects for future development trends. We have revised the paper. Highlighted the elements of the novelty in the part of abstract, introduction, conclusion and future perspectives. Please see line 14-33, 123-130, and 822-867.

Point 2: The authors are encouraged to indicate their contribution to the development of AUV trajectory tracking models and control strategies.

Response 2: Thanks for your constructive comments. Following your comments, we have revised this paper. We indicated our contribution to the development of AUV trajectory tracking models and control strategies in detail. We have mainly revised the abstract, introduction, and conclusion.

In abstract, we pointed out two methods of modeling and three aspects about control strategies that need to be considered in the control process. We introduced that in the aspect of control strategies, mathematical modeling study and physical experiment study are introduced in detail. And with the aim of establishing the acceptability of the reported modeling and control techniques, as well as challenges that remain open, a discussion and a case study are presented. We point out that the development of new control-oriented models, the research in the estimation of un-known inputs, and the development of more innovative control strategies for AUV trajectory tracking systems are still open problems that must be addressed in the short term. Please see line 14-33.

In introduction, we introduced AUV and AUV trajectory tracking control. The latest AUV name, research organization and research purpose of different countries have been introduced. Which could be introduced for researchers to conduct deeper study. We emphasised on the challenges in AUV modeling and control, and we have a lot of discussion about the modeling methods and control strategies. Furthermore, we proposed the future development of modeling methods and control strategies. Please see line 37-130.

In conclusion and future perspectives, we emphasized the status quo of the development of modeling and control. At the same time, the advantages of deep learning and reinforcement learning in data processing will be of great significance in the field of AUV trajectory tracking modeling and control. Then three case studies were introduced, focusing on the experimental equipment, software used, experimental constraints, and experimental results. At the same time, we compared the existing control strategies and proposed that the combination of neural network and traditional algorithms has achieved important results in AUV trajectory tracking. Finally, we propose that the development of new control-oriented models, the research in the estimation of un-known inputs, and the development of more innovative control strategies for AUV trajectory tracking systems are still open problems that must be addressed in the short term. Please see 822-867.

Point 3:  Figure 1 seems to be insufficiently described.

Response: Thanks for your constructive comments. Following your comments, we describe the Figure 1 in detail. The structure and principle of AUV trajectory tracking control are introduced in detail, so that readers can better understand AUV trajectory tracking control. The specific modifications are as follows:

The control principle of AUV trajectory tracking is shown in Fig. 1. AUV trajectory tracking control includes trajectory planning, user interface, and trajectory tracking control three parts. The trajectory planning includes path planning, behaviour decision, and trajectory generation. It generates by the required database such as task and mode data, vehicle component, and user profile. The GUI displays interactive data, including trajectory planning and trajectory control two aspects. AUV obtains the planned route through the GUI, and it generates deviation with the actual movement data, and the motion control is achieved through thruster control and energy control, thereby the trajectory tracking error is reduced. Please see line 64-72.

Point 4:  Please mark the vectors in bold.

Response: Thanks for your constructive comments. Following your comments, we carefully checked the symbols in the article, especially the equations (1) ~ (16). The vectors in the equations are expressed in bold. The revisions are highlighted in yellow in the revised manuscript.

Point 5:  The References should be extended to include the publications on intelligent solutions in shipping that refer to the discussed subject, for instance:

Qu, Y.; Xiao, B.; Fu, Z.; Yuan, D. Trajectory exponential tracking control of unmanned surface ships with external disturbance and system uncertainties. ISA Transactions 2018, 78, 47-55.

Borkowski P., Zwierzewicz Z. “Ship course-keeping algorithm based on knowledge base” Intelligent Automation and Soft Computing vol. 17, no. 2, 2011 (149-163)

Response: Thanks for your constructive comments. Following your comments, we have carefully read these related references that are the good papers with high quality and new results. These two papers indeed help us improve the quality of our paper especially for the methodology and applications. Meanwhile, we have cited these two papers. The specific modifications are as follows:

Although several challenges as stated above have been addressed, external disturbance and system uncertainties are another two issues that need to be solved in tracking controller design for surface ships. To solve these problems, an observer-based estimation was preliminarily designed by [106] to estimate the disturbance and system uncertainties. Then, a backstepping controller was synthesized with a compensation control effort incorporated for accommodating the disturbance and uncertainties. The proposed controller guaranteed that the desired trajectory can be followed with an exponential rate of convergence. Please see line 545-552.

Based on a knowledge base, [129] presented an original ship course-keeping algorithm. Its integral part is a computer-borne ship movement dynamical model based on a set of signals obtained from the object’s input and output, which has been avoided the problems occurring while designing classic control algorithms for a complex, non-linear ship model. The designed algorithm was compared to LQR controller as well as feedback linearizing one. The results prove high quality performance of the proposed method. Please see line 643-649.

We have colored the related changes in yellow in the revised manuscript for your convenience. Please view the changes in revised manuscript. We would like to thank you again for your valuable comments that truly help us improve this paper for the final publication.

Sincerely,

Daoliang Li

Ling Du